# Q*: Improving Multi-step Reasoning for LLMs with Deliberative Planning

## Abstract

Large Language Models (LLMs) have demonstrated impressive capability across various natural language tasks. However, the auto-regressive generation process makes LLMs prone to produce errors, hallucinations and inconsistent statements when performing multi-step reasoning. In this paper, by casting multi-step reasoning of LLMs as a heuristic search problem, we aim to alleviate the pathology by introducing Q*, a general, versatile and agile framework for guiding LLMs decoding process with deliberative planning. By learning a plug-and-play Q-value model as heuristic function for estimating expected future rewards, Q* can effectively guide LLMs to select the most promising next reasoning step without fine-tuning LLMs for the targeted task, which avoids the significant computational overhead and potential risk of performance degeneration on other tasks. Extensive experiments on GSM8K, MATH and MBPP datasets demonstrate the superiority of our method, contributing to improving the reasoning capability of existing open-source LLMs. Furthermore, the testing-time scaling law indicates that Q* can leverage increased computational power to improve reasoning performance.

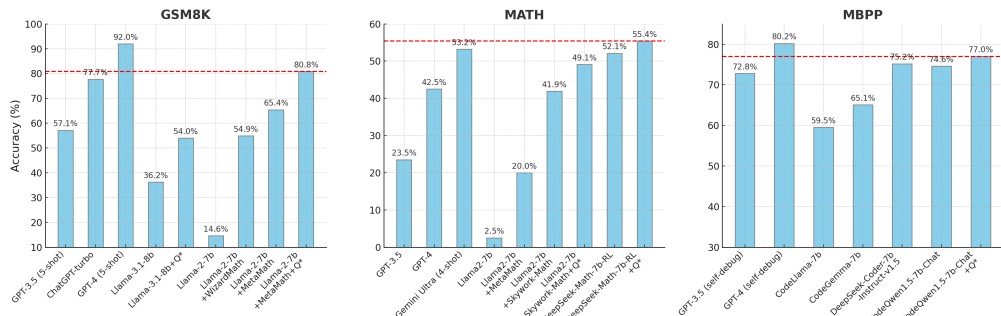

Figure 1: Performance comparison of Q* with other baselines. Q* can serve as an efficient testing-time alignment technique which significantly improves the performance of open-source LLMs on math reasoning tasks (GSM8K: **36.2%→54.0%**; **65.4%→80.8%**, MATH: **52.1%→55.4%**) and code generation task (MBPP: **74.6%→77.0%**) without modifying the model's parameters.

## 1 Introduction

Large Language Models (LLMs) have exhibited impressive capabilities in solving various reasoning tasks encoded in natural languages, including math word problems (Ahn et al., 2024; Cobbe et al., 2021; Hendrycks et al., 2021; Wang et al., 2023; Yu et al., 2023; Luo et al., 2023a), code generation (Luo et al., 2023b; Roziere et al., 2023; CodeGemma Team et al., 2024) and planning (Xie et al., 2024; Liu et al., 2023; Guan et al., 2023). Unfortunately, even the most advanced LLMs still face significant challenges and are prone to introduce errors, hallucinations and inconsistent statements as the number of reasoning steps grows due to their auto-regressive nature (Valmeekam et al., 2023; Stechly et al., 2024). In fact, the auto-regressive generation process of LLMs can be characterized by "System 1" (Daniel, 2011), a mode of thought which is fast, instinctive but less accurate. Most of recent works focus on improving LLMs' "System 1" capability by (1) constructing sophisticated prompts with

extensive expertise to trigger the potential capacities of LLMs without modifying their parameters (Wei et al., 2022; Wang et al., 2022; Fu et al., 2022; Zhou et al., 2022), (2) fine-tuning LLMs with massive task-specific corpora at the price of significant computational burdens and the potential risk of performance degeneration on other tasks (Yu et al., 2023; Luo et al., 2023a; Azerbayev et al., 2023; Yue et al., 2023), or (3) training reward models to rank the candidate responses (Lightman et al., 2023; Uesato et al., 2022; Wang et al., 2023; Khalifa et al., 2023).

On the other hand, solving complex reasoning problems requires more in-depth, deliberative and logical thinking steps, *i.e.*, the "System 2" mode (Daniel, 2011). Taking solving math word problems as an example, any incorrect intermediate reasoning step (*e.g.*, calculation errors, mis-interpretations) can potentially lead to incorrect final answers. Prior attempts (Yao et al., 2023; Feng et al., 2023; Hao et al., 2023; Zhuang et al., 2023) for enhancing "System 2" reasoning capability includes performing deliberation with basic tree search algorithms (*e.g.*, BFS or DFS), Monte Carlo Tree Search (MCTS) (Browne et al., 2012), and A* (Hart et al., 1968). Nonetheless, the utility functions used in these methods often require laborious expertise to design for each specific task, which are difficult to be extended to new scenarios. Furthermore, deliberation with MCTS would require significant number of rollouts before finding high-quality responses when solving the problems with many reasoning steps, which substantially slows down the overall decoding process. Very recently, OpenAI released its o1 series (OpenAI, 2024), an LLM capable of solving complex reasoning tasks by leveraging increased computational resources at the inference time to achieve better problem-solving performance. Unfortunately, as a propertied model, it is still unclear how o1 produces the long internal chain-of-thought which is the key of its superior performance.

In light of this, we propose Q*, a general, versatile and agile framework for improving the multi-step reasoning capability of LLMs with deliberative planning. Different from the existing deliberation methods, our method does not rely on domain knowledge to design the heuristic function. Besides, by leveraging plug-and-play Q-value models as heuristic function, Q* can effectively solve various tasks via guiding LLMs to select the most promising next step without fine-tuning LLMs beforehand, which avoids the significant computational overhead and potential risk of performance degeneration in other tasks. Finally, Q* considers only one single step when performing deliberation, which is much cheaper than completing rollouts in MCTS. In short, Q* can serve as an efficient testing-time alignment technique for LLMs which consistently improves the performance on various complex reasoning tasks, as evidenced by Fig. 1. Specifically, the main contributions of our work are summarized as follows:

- We formalize the multi-step reasoning of LLMs as a Markov Decision Process (MDP) where the state is the concatenation of input prompt and the reasoning steps generated so far, the action is the next reasoning step and the reward measures how well the task is solved.

- We present several general approaches to estimate the optimal Q-value of state-action pairs, *i.e.*, offline reinforcement learning, best-of-$K$ sampling and MCTS planning. It is noteworthy that our methods only need the ground-truth of training problems and can be flexibly applied to various reasoning tasks without modification.

- We cast solving multi-step reasoning tasks as a heuristic search problem, where the objective is to find the most proper reasoning trace with maximum utility. Built upon A* search, our deliberation framework, Q*, leverages plug-and-play Q-value models as heuristic function and guides LLMs to select the most promising next reasoning step in best-first fashion.

- We conduct extensive experiments on math reasoning and code generation tasks, which demonstrates that Q* can significantly improve the multi-step reasoning capability of existing open-source LLMs. Furthermore, the testing-time scaling law of Q* exhibits performance improvement over generated tokens, indicating that Q* can continuously refine its solution with the increased computational cost.

## 2 RELATED WORKS

**Alignment in LLMs.** Alignment has become an important technique to prevent the output of LLMs deviates from human's expectation. Supervised Fine-Tuning (SFT) is probably the most fundamental alignment approach that directly minimizes the cross-entropy loss between the output and ground-truth. Reinforcement learning from Human Feedback (RLHF) (Ouyang et al., 2022),

on the other hand, firstly learns a reward model (RM) from human preferences and then optimizes the SFT model with reinforcement learning algorithms to maximize the cumulative rewards from RM. Direct Preference Optimization (DPO) (Rafailov et al., 2023) aligns LLMs directly according to the ranking information from human feedback without explicitly learning RM. Recently, Aligner (Ji et al., 2024) came out as a model-agnostic alignment method by learning to re-write LLMs' output. Compared to these methods, Q* achieves the goal of alignment with distinct merits. Different from SFT and Aligner, Q* does not rely on massive human annotated preference pairs which are expensive to collect; different from RLHF and DPO, Q* does not modify the parameters of LLMs, which avoids the potential risk of performance degeneration on other tasks. In short, Q* can serve as an efficient testing-time alignment technique by searching the most proper chain-of-thought for a given reasoning task.

**Enhancing LLMs with planning.** Tree-of-thoughts (ToT) (Yao et al., 2023) improves the LLMs' reasoning capability by exploring the intermediate steps towards problem solving with basic tree-search algorithms. In the same vein, A* search and MCTS have been applied to serve as a planning technique to enhance the performance of LLMs when solving challenging complex reasoning problems (Feng et al., 2023; Hao et al., 2023; Zhuang et al., 2023; Hazra et al., 2024). Unfortunately, the utility function used in these methods is either constructed from LLMs' feedback (*e.g.*, Yao et al. (2023); Hao et al. (2023)), which could be highly-inaccurate in complex problems, or specific to each individual task (*e.g.*, Zhuang et al. (2023); Hazra et al. (2024)). Moreover, planning with MCTS often requires to perform costly rollout, which can significantly slow down the overall decoding process. In contrast, Q* solely relies on training a Q-value model to guide LLMs to select the most promising next reasoning step and the pipeline can be easily applied to various reasoning tasks without modification. Besides, we consider only a single step each time in Q*, which is much cheaper than a complete rollout in common MCTS-based methods.

**LLMs for math reasoning & code generation.** Math reasoning and code generation require LLMs to perform multi-step reasoning on relations, quantities and logics which are inherently challenging. Current techniques include: 1) prompt engineering which triggers the potential capacities of LLMs with sophisticated prompts (Wei et al., 2022; Wang et al., 2022; Fu et al., 2022; Zhou et al., 2022; Huang et al., 2023; Shinn et al., 2023). However, constructing such prompt needs extensive expertise and case-by-case tuning, which is difficult to generalize to different tasks; 2) Fine-tuning LLMs with massive math/code corpus (Yu et al., 2023; Luo et al., 2023a; Azerbayev et al., 2023; Yue et al., 2023; Roziere et al., 2023; CodeGemma Team et al., 2024; Team, 2024), which usually comes at the price of significant computational burden and may compromise the performance on other tasks; 3) training RMs/verifiers to rank the candidate solutions without providing any guidance in intermediate steps (Lightman et al., 2023; Uesato et al., 2022; Wang et al., 2023; Khalifa et al., 2023). Differently, Q* leverages a plug-and-play Q-value model to direct the deliberation process of LLMs, which effectively provides guidance for each intermediate step without modifying the parameters of LLMs. Moreover, by casting multi-step reasoning of LLMs as a heuristic search problem, our method can be generalized to various reasoning tasks without laborious prompt engineering. Besides, OpenAI's o1 (OpenAI, 2024) demonstrates its superior performance in various tasks including math reasoning and code generation. However, as a propertied model, it is unclear how o1 generates the long internal chain-of-thought which is essential to successfully solving a problem. In contrast, Q* provides an alternative yet efficient way to implement testing-time deliberation for LLMs, and we will release codes if the paper is accepted.

## 3 PRELIMINARY

### 3.1 FORMULATE THE MULTI-STEP REASONING OF LLMs AS AN MDP

Taking the question $q$ as input, the answer generation process of LLMs can be broken down into multiple reasoning steps, where the final answer sequence $\mathbf{a}$ can be treated as the concatenation of these $T$ single-step reasoning steps, formulated as $\mathbf{a} = [a_1; a_2; \ldots; a_T]$. Each step $a_t$ can be a single line or fixed number of tokens outputted by LLMs. Under this perspective, we can conceptualize the multi-step reasoning process of LLMs as a Markov Decision Process (MDP) $\langle \mathcal{S}, \mathcal{A}, \mathcal{T}, \mathcal{R}, \gamma \rangle$, where the state $s_t \in \mathcal{S}$ denotes the concatenation of the input question and the partial reasoning trace already generated by timestep $t - 1$ (*i.e.*, $s_t = [q; a_1; \ldots; a_{t-1}]$) with the special definition

$s_1 = q$, the action $a_t \in \mathcal{A}$ denotes the next reasoning step generated by LLMs taking the current state $s_t$ as input, the deterministic state transition $\mathcal{T}$ from the current state $s_t$ to the next state $s_{t+1}$ is accomplished through a simple operation of concatenation, $\mathcal{R}$ is the reward function to measure how well the question is solved and $\gamma$ is the discount factor. The reward function is often *outcome-based*. That is, it gives reward by comparing the final results with ground-truth:

$$\mathcal{R}(s_t, a_t) = \begin{cases} 1 & t = T \wedge [s_t; a_t] \text{ matches the ground-truth} \\ 0 & \text{otherwise} \end{cases}, \tag{1}$$

In particular, we will assign a reward of 1 if the generated code passes all test cases (for code generation tasks) or the final answer matches the ground-truth (for math reasoning tasks), which is a common practise in previous studies (Wang et al., 2023; Lightman et al., 2023). Finally, the policy $\pi_\theta$ is embodied by an LLM, which produces reasoning sequence conditioned on the input question:

$$\pi_\theta(a_t|s_t) = \text{LLM}(a_t|s_t), \ \pi_\theta(\mathbf{a}|q) = \prod_{t=1}^{T} \pi_\theta(a_t|s_t). \tag{2}$$

Given the MDP and LLM policy $\pi_\theta$, the *value* of state-action pair $(s_t, a_t)$ is given by a *Q-function* $Q^{\pi_\theta}(s_t, a_t) = \mathbb{E}_{\pi_\theta} \left[ \sum_{t'=t}^{T} \gamma^{T-t'} \mathcal{R}(s_{t'}, a_{t'}) \right]$. The Q-function of an optimal policy $\pi^*$ is called *optimal Q-function* and satisfies the Bellman optimality equation:

$$Q^*(s_t, a_t) = \mathcal{R}(s_t, a_t) + \gamma \max_{a_{t+1} \in \mathcal{A}} Q^*(s_{t+1}, a_{t+1}), \tag{3}$$

which gives the value of starting state $s_t$, taking action $a_t$ and then following the optimal policy $\pi^*$.

## 3.2 A* Search

**A\*** (Hart et al., 1968) is an important heuristic search algorithm in deliberative planning (Bonet & Geffner, 2001), multi-agent pathfinding (Silver, 2005), and constraint reasoning (Pezeshki et al., 2022). Originally, A* is proposed for finding the shortest path from source $s$ to goal $g$ in path planning problems. It associates each frontier vertex $n$ with a value $f(n) = g(n) + h(n)$, where $g(n)$ is the accumulated path cost from source $s$ and $h(n)$ is a heuristic value that estimates the cost of the shortest path from $n$ to goal $g$. The algorithm adopts a best-first search strategy, *i.e.*, in each iteration it always picks the vertex with minimum $f$-value to explore until reaching the goal. When the heuristic $h(\cdot)$ is *admissible* (Russell & Norvig, 2016), A* guarantees to find the optimal path.

# 4 Q*: A GENERAL, VERSATILE AND AGILE DELIBERATION FRAMEWORK FOR LLMS

Most of modern LLMs generate natural languages in an auto-regressive way, *i.e.*, predict the next token in a sequence given the previously generated tokens (cf. Eq. (2)). Therefore, when applied to multi-step reasoning problem, LLMs can potentially introduce errors, hallucinations and inconsistent statements in the subsequent reasoning trace if any previous step is incorrect, which may fail to solve the current problem. Indeed, given the fact that LLMs produce each token with limited computation resources, there is no way to devote more computational efforts to solve difficult problems. In short, LLMs cannot perform in-depth deliberation which is essential for solving complex multi-step reasoning problems.

We address this issue by presenting Q*, a general, versatile and agile deliberation framework based on A* to effectively guide LLMs to select the most promising next step when performing multi-step reasoning without costly fine-tuning LLMs for each task beforehand. In more detail, we cast finding the most proper reasoning sequence for a given problem as a heuristic search process, where each state $s_t$ is associated with a $f$-value estimating how much utility will be attained if we expand $s_t$:

$$f(s_t) = g(s_t) + \lambda h(s_t), \tag{4}$$

where $g(s_t)$ denotes the aggregated utility from the initial state $s_1$; $h(s_t)$ is the heuristic value for measuring the probability of reaching the correct answer derived from $s_t$; $\lambda$ is a coefficient to balance the importance of $g(s_t)$ and $h(s_t)$ terms.

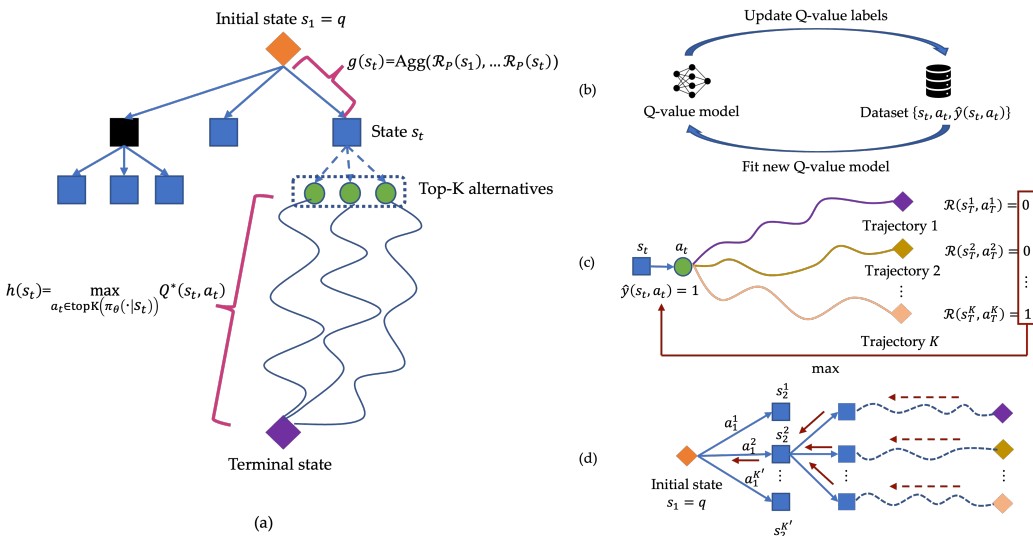

Figure 2: Overview of Q*. **(a)**: the deliberation process of Q*. Each state is associated with an $f$-value which is the weighted sum of the aggregated utility (cf. Eq. (5)) and the heuristic value (cf. Eq. (6)). **(b-d)**: estimating the optimal Q-value with offline reinforcement learning, best-of-$K$ sampling and MCTS planning.

Specifically, we propose to use process-based reward function $\mathcal{R}_P$ that encodes the prior knowledge or preference of the reasoning task to compute the aggregated utility $g(s_t)$. That is,

$$g(s_t) = \text{Agg}(\mathcal{R}_P(s_1), \ldots, \mathcal{R}_P(s_i), \ldots, \mathcal{R}_P(s_t)), \tag{5}$$

where $\text{Agg} \in \{\min, \max, \sum, [-1]\}$, with $[-1]$ standing for assigning the reward of last state as the utility, is the aggregation function to summarize the rewards in the path from $s_1$ to $s_t$, and $s_{i-1}$ is the prefix of $s_i$, $1 < i \leq t$. Such process-based reward function $\mathcal{R}_P$ could be learned from human feedback (Lightman et al., 2023; Uesato et al., 2022; Wu et al., 2023), ground-truth (Wang et al., 2023; Khalifa et al., 2023), rules, or simply be the logits of a reasoning step which reflects the confidence of the LLM. Furthermore, we use the optimal Q-value of state $s_t$ (cf. Eq. (3)) as the heuristic value $h(s_t)$. In other words, the $f$-value is given by:

$$f(s_t) = g(s_t) + \lambda \max_{a_t \in \mathcal{A}} Q^*(s_t, a_t). \tag{6}$$

Since enumerating all possible next reasoning steps is intractable, in practice one can restrict the alternatives to the top-K of all step candidates returned by LLM, and thus Eq. (6) is written as $f(s_t) = g(s_t) + \lambda \max_{a_t \in \text{top-K}(\pi_\theta(\cdot|s_t))} Q^*(s_t, a_t)$.

## 4.1 ESTIMATION OF OPTIMAL Q-VALUE

A critical challenge of implementing Q* is to estimate the optimal Q-value of state-action pairs (cf. Eq. (6)) with a frozen LLM policy $\pi_\theta$ which could be suboptimal on the given reasoning problems. Specifically, we aim to learn a proxy Q-value model $\hat{Q}$ to approximate $Q^*$ from a dataset $D = \{q_i, \{\mathbf{a}_i^{(j)}\}_{j=1}^M\}_{i=1}^N$, where $q_i$ is a training problem and $\mathbf{a}_i^{(j)}$ is the $j$-th trajectory sampled from the LLM policy $\pi_\theta$ with a particular technique. Formally:

$$\hat{Q} = \arg\min_Q \frac{1}{NMT} \sum_{i=1}^N \sum_{j=1}^M \sum_{a_t \in \mathbf{a}_i^{(j)}} \left(Q(s_t, a_t) - \hat{y}(s_t, a_t)\right)^2, \tag{7}$$

where $s_t = [q_i; a_1; \ldots; a_{t-1}]$ is the partial reasoning trace by timestep $t-1$ in $\mathbf{a}_i^{(j)}$ and $\hat{y}(s_t, a_t)$ is the label that approximates the true optimal Q-value, specifically $Q^*(s_t, a_t)$.

In more detail, we effectively construct the dataset $D$ and Q-value labels $\hat{y}(s_t, a_t)$ for question $q_i$ in the following ways.

**Offline reinforcement learning.** For each training problem $q_i$, we directly sample $M$ reasoning trajectories $\{\mathbf{a}_i^{(j)}\}_{j=1}^M$ from the LLM policy, where each trajectory $\mathbf{a}_i^{(j)} \sim \pi_\theta(\cdot|q_i)$. After that, we learn the proxy Q-value model $\hat{Q}$ using Fitted Q-iteration (Riedmiller, 2005). Specifically, for each iteration $\ell$, we construct Q-value label as:

$$\hat{y}_\ell(s_t, a_t) = \begin{cases} \mathcal{R}(s_t, a_t) & t = T \\ \mathcal{R}(s_t, a_t) + \gamma \max_{a_{t+1}\in\text{top-K}(\pi_\theta(\cdot|s_{t+1}))} \hat{Q}_{\ell-1}(s_{t+1}, a_{t+1}) & \text{otherwise} \end{cases}, \quad (8)$$

where $\hat{Q}_{\ell-1}$ is the proxy Q-value model learned in iteration $\ell - 1$. Then, we train a new proxy model $\hat{Q}_\ell$ according to Eq. (7). Such two phases will be alternated for $L$ iterations, and we use $\hat{Q}_L$ as the proxy Q-value model when performing deliberation.

**Best-of-$K$ sampling.** Similar to offline RL, we will firstly construct the dataset $D$ by randomly rolling out trajectories with $\pi_\theta$. Then starting with each state-action pair $(s_t, a_t)$ in a trajectory $\mathbf{a}_i^{(j)}$, we perform random sampling with the LLM policy $\pi_\theta$ to complete it into $K$ full trajectories $\{\tau_k\}_{k=1}^K$, where $\tau_k \sim \pi_\theta(\cdot|[s_t; a_t])$. After that, we use the best reasoning trajectory with the highest accumulated rewards to construct the Q-value label:

$$\hat{y}(s_t, a_t) = \mathcal{R}(s_t, a_t) + \max_{(s_{t'}, a_{t'})\in\tau_k} \left[ \sum_{t'=t+1}^T \gamma^{T-t'} \mathcal{R}(s_{t'}, a_{t'}) \right]. \quad (9)$$

**MCTS planning.** For each training problem $q_i$, we perform canonical MCTS (Browne et al., 2012) to collect reasoning trajectories $\{\mathbf{a}_i^{(j)}\}_{j=1}^M$ and the corresponding Q-value labels. Starting from $s_1 = q_i$, we incrementally build a search tree $\Gamma_i$ in which each node and edge respectively correspond to a state and an action through four phases: (1) **Selection.** recursively selecting the most promising child node until leaf node with UCB1 bound (Auer et al., 2002); (2) **Expansion.** sampling $K'$ different next reasoning steps using the LLM policy $\pi_\theta$ and generating $K'$ new child nodes on top of the leaf node; (3) **Simulation.** performing rollout from the new nodes with the LLM policy $\pi_\theta$ until terminal states to produce complete trajectories; (4) **Backpropagation.** updating the value of each edge in the path from the root to the leaf node with the reward of the trajectory (cf. Eq. (1)). Finally, we retrieve reasoning trajectories $\{\mathbf{a}_i^{(j)}\}_{j=1}^M$ by performing depth-first search on $\Gamma_i$ to collect all paths from the root to the terminal nodes, and use the value of each edge as Q-value label $\hat{y}(s_t, a_t)$.

### 4.2 Deliberative Planning with A*

Once obtaining the proxy Q-value model $\hat{Q}$, we can plug it to Eq. (6) to compute the $f$-value of each state and perform best-first search with A*. Alg. 1 illustrates the deliberative planning process.

---

**Algorithm 1** Deliberative planning for LLMs with A*

---

**Input:** question $q$, LLM policy $\pi_\theta$, proxy Q-value model $\hat{Q}$
**Output:** best reasoning trajectory $s^*$
 1: $unvisited \leftarrow \{q\}$, $visited \leftarrow \emptyset$, terminal states $S_T \leftarrow \emptyset$
 2: **while** $unvisited \neq \emptyset$ and termination condition is not met **do**
 3:    $s \leftarrow \arg\max_{s'\in unvisited} f(s')$
 4:    $unvisited \leftarrow unvisited\backslash\{s\}$, $visited \leftarrow visited \cup \{s\}$
 5:    **if** $s$ is a terminal state **then**
 6:       $S_T \leftarrow S_T \cup \{s\}$
 7:       **continue**
 8:    **for each** $a \in \text{top-K}(\pi_\theta(\cdot|s))$ **do**
 9:       $s' \leftarrow [s; a]$
10:       **if** $s' \notin visited$ **then** $unvisited \leftarrow unvisited \cup \{s'\}$
11: $s^* \leftarrow \arg\max_{s'\in S_T} f(s')$
12: **return** $s^*$

---

Specifically, we maintain a set for storing state candidates to be explored, denoted as $unvisited$, which initially only contains the input question $q$, and another set $visited$ to record the visited states.

Each step we pick the state $s$ with the maximum $f$-value from the set $unvisited$ and expand it by querying the top-K alternatives with the LLM policy $\pi_\theta$ if it is not a terminal state (i.e., a complete reasoning trajectory). After that, both $visited$ and $unvisited$ sets will be updated and this process repeats until the termination condition is met or all states are visited. Finally, Q* will return the best reasoning trajectory $s^* = [q; a_1^*; \ldots; a_T^*]$ among the set of collected terminal states, denoted as $S_T$, as the final result.

## 5 EXPERIMENTS

### 5.1 EXPERIMENTAL SETTINGS

**Datasets.** We evaluate the effectiveness of Q* on two math reasoning and one code generation tasks, where the dataset statistics have been summarized in Table 1. 1) GSM8K (Cobbe et al., 2021) is a dataset of grade school math problems, where the solution is given in a one-line-per-step format with an exact numerical answer in the last line; 2) MATH (Hendrycks et al., 2021) is a dataset consisting of math problems of high school math competitions, where the solutions are given in a format that mixes latex code and natural language; 3) MBPP (Austin et al., 2021) is an entry-level Python programming dataset, where the questions are coding challenges along with a test case that defines the function format. The solutions are Python code that is excepted to pass the pre-collected test cases.

Table 1: Statistics of datasets.

| Dataset | GSM8K | MATH | MBPP |
|---|---|---|---|
| Domain | Math Reasoning | Math Reasoning | Code Generation |
| Training | 5000 | 8000 | 374 |
| Testing | 1319 | 5000 | 500 |
| Average Steps | 4.5 | 11.0 | 7.0 |

Table 2: Comparisons of different methods for Q-value estimation.

| Dataset (Domain) | Base Model | Q-value Estimation | Planning | Accuracy |
|---|---|---|---|---|
| GSM8K (Math Reasoning) | Llama-2-7b-MetaMath | Offline RL | Q* | 68.2% |
| GSM8K (Math Reasoning) | Llama-2-7b-MetaMath | Best-of-$K$ | Q* | **79.8**% |
| GSM8K (Math Reasoning) | Llama-2-7b-MetaMath | MCTS planning | Q* | 70.1% |
| MBPP (Code Generation) | CodeQwen1.5-7b-Chat | Offline RL | Q* | 76.4% |
| MBPP (Code Generation) | CodeQwen1.5-7b-Chat | Best-of-$K$ | Q* | **77.0**% |
| MBPP (Code Generation) | CodeQwen1.5-7b-Chat | MCTS planning | Q* | 76.6% |

**Implementation Details.** The implementation of Q* method mainly includes three steps:

1) ***Q-value estimation.*** As discussed in Section 4.1, we propose several ways for estimating the optimal Q-values, and by comparing the performance of these estimation methods, as shown in the Table. 2, we find that best-of-$K$ sampling could be the most effective and robust way to collect precise Q-value labels. Specifically, for each training question $q_i$, we will firstly perform sampling to obtain $M = 160$ complete trajectories $\{\mathbf{a}_i^{(j)}\}_{j=1}^M$ with the LLM policy $\pi_\theta$, under the setting of high temperature, *e.g.*, $\tau = 0.9$ for math reasoning and $\tau = 0.2$ for code generation, and split each trajectory into a series of step-level states according to the newline token "$\backslash n$". Then, for each state-action pair in a trajectory, denoted as $(s_t, a_t)$, we perform best-of-$K$ sampling with the same LLM to generate complete trajectories $\{\tau_k\}_{k=1}^K$ where $K = 16$, and then select the best reasoning trajectory with the highest accumulated rewards as the Q-value label of the current state-action pair. Besides, we use $\gamma = 1$ as the discount factor. Therefore, the optimal Q-value of a state-action pair $(s_t, a_t)$ will be assigned as 1 if and only if it has the potential to generate a trajectory that matches the ground-truth, *i.e.,* the answer is correct in math reasoning or the code program can pass all test cases in code generation. Finally, we will initialize the Q-value models (QVMs) using the same base model as the LLM policy $\pi_\theta$ and train them as regressors to approximate the optimal Q-values.

2) ***Utility aggregation.*** For GSM8K dataset, we adopt a process reward model (PRM) trained on PRM800K (Lightman et al., 2023) to model $\mathcal{R}_P$ to provide an intermediate signal for each reasoning step, and use $\min$ as the aggregation function; For MATH dataset, we set $g(s_t) = 0$ for all passed states $\{s_i\}_{i=1}^t$ in each trajectory for fairness, because PRM800K contains data samples constructed

Table 3: Comparison of Q* and other baselines on GSM8K dataset.

| Base Model | SFT | Post-Training | Planning | Accuracy |
|---|---|---|---|---|
| GPT-3.5 (5-shot) (Achiam et al., 2023) | Unknown | PPO (RM) (Ouyang et al., 2022) | - | 57.1% |
| ChatGPT-instruct (0-shot) (Shridhar et al., 2023) | Unknown | PPO (RM) (Ouyang et al., 2022) | - | 71.3% |
| ChatGPT-turbo (0-shot) (Shridhar et al., 2023) | Unknown | PPO (RM) (Ouyang et al., 2022) | - | 77.7% |
| GPT-4 (0-shot) (Shridhar et al., 2023) | Unknown | PPO (RM) (Ouyang et al., 2022) | - | 91.9% |
| GPT-4 (5-shot) (Achiam et al., 2023) | Unknown | PPO (RM) (Ouyang et al., 2022) | - | **92.0%** |
| Llama-2-7b (0-shot) | - | - | - | 14.6% |
| Llama-2-7b (0-shot) | WizardMath(Luo et al., 2023a) | - | - | 54.9% |
| Llama-2-7b (0-shot) | MetaMath(Yu et al., 2023) | - | - | 65.4% |
| Llama-2-7b (0-shot) | MetaMath(Yu et al., 2023) | PPO (PRM) Ouyang et al. (2022) | - | 67.2% |
| Llama-2-7b (0-shot) | MetaMath(Yu et al., 2023) | PPO (QVM) Ouyang et al. (2022) | - | 67.6% |
| Llama-2-7b (0-shot) | MetaMath(Yu et al., 2023) | - | Best-of-$N$ (PRM) | 72.1% |
| Llama-2-7b (0-shot) | MetaMath(Yu et al., 2023) | - | Best-of-$N$ (QVM) | 74.5% |
| Llama-2-7b (0-shot) | MetaMath(Yu et al., 2023) | - | MCTS (QVM) | 77.6% |
| Llama-2-7b (0-shot) | MetaMath(Yu et al., 2023) | - | Q* (QVM) | 79.8% |
| Llama-2-7b (0-shot) | MetaMath(Yu et al., 2023) | - | Q* (PRM+QVM) | **80.8%** |

from MATH testing set and there is a potential risk of data leakage; For MBPP dataset, we tokenize the code generated so far with function `tokenize.generate_tokens` and give a penalty of -0.5 if `TokenError` is raised, which is often the case that there are mismatched delimiters (*e.g.*, parentheses, quotation marks) and invalid indention in the code. We use $[-1]$ as the aggregation function to cancel the previous penalties since the code is generated on-the-fly and mismatched delimiters may be fixed in subsequent steps.

3) *A* planning.* For GSM8K and MATH datasets, we treat a single line outputted by the LLM as an action, while the action in MBPP dataset is defined as a code snippet with 24 tokens when planning. Besides, when computing the $f$-values defined in Eq. (6) in all experiments, we set $\lambda = 1$ and expand each state with $K = 6$ actions at each reasoning step. Finally, following the common practice of Best-of-$N$ (Lightman et al., 2023), we perform planning to collect $N = 6$ trajectories for each question, and select the one with the maximum $f$-value as the final result for evaluation.

## 5.2 ESTIMATIONS OF OPTIMAL Q-VALUE

We first compare the performance of QVMs trained with three different approaches presented in Section 4.1. Specifically, for offline RL and best-of-$K$ sampling, we first collect 64 positive trajectories (*i.e.*, the answer is correct or the code program passes all test cases) and 96 negative trajectories for each training question $q_i$. Then for each state-action pair in a trajectory, we perform $L = 6$ iterations of fitted-Q-iteration or $K = 16$ sampling to construct Q-value labels. Finally, for MCTS planning, we perform 1024 iterations of selection-expand-simulation-backpropagation for each training question $q_i$, and extract Q-value labels from the search tree $\Gamma_i$. Table 2 displays the performance of Q* with the QVMs trained with different methods on GSM8K and MBPP datasets.

By approximating with the best response among $K$ sampling with current LLM policy $\pi_\theta$, best-of-$K$ sampling emerges as a simple yet effective method for estimating optimal Q-values, achieving superior performance on both datasets. In fact, LLMs have the potential to generate correct answer given a generous rollout budget (Li et al., 2024). Therefore, selecting the best response according to the ground-truth can effectively approximate the behavior of optimal policy, as well as the optimal Q-values. Offline RL, on the other hand, exhibits inferior performance because it *indirectly* learns Q-values by querying intermediate QVMs, which is inefficient and may be biased by QVMs and top-K alternatives (cf. Eq.(8)). Finally, MCTS planning receives feedback from the ground-truth, directing the search to find promising trajectories. As a result, the Q-value labels in the generated dataset is imbalanced, with the majority being close to 1, which can also bias the learning of QVMs.

## 5.3 QUANTITATIVE COMPARISON

**GSM8K.** For the comparison on GSM8K dataset, we select Llama-2-7b (Touvron et al., 2023) as our base model, whose accuracy can achieve 65.2% after finetuning on MetaMath (Yu et al., 2023). Then, we treat Llama-2-7b finetuned on MetaMath as the LLM policy $\pi_\theta$, and perform best-of-$K$ sampling to collect Q-value labels for training QVM. For utility aggregation, we train a process reward model (PRM) on PRM800K (Lightman et al., 2023) to provide intermediate signal for each

Table 4: Comparison of Q* and other baselines on MATH dataset.

| Base Model | SFT | Post-training | Planning | Accuracy |
|---|---|---|---|---|
| GPT-3.5 (0-shot) (Bubeck et al., 2023) | Unknown | PPO (RM) (Ouyang et al., 2022) | - | 23.5% |
| GPT-4 (0-shot) (Bubeck et al., 2023) | Unknown | PPO (RM) (Ouyang et al., 2022) | - | 42.5% |
| Gemini Ultra (4-shot) (Team et al., 2023) | Unknown | PPO (RM) (Ouyang et al., 2022) | - | **53.2**% |
| Llama-2-7b (0-shot) | - | - | - | 2.5% |
| Llama-2-7b (0-shot) | MetaMath(Yu et al., 2023) | - | - | 20.0% |
| Llama-2-7b (0-shot) | Skywork-Math(Zeng et al., 2024) | - | - | 41.9% |
| Llama-2-7b (0-shot) | Skywork-Math(Zeng et al., 2024) | PPO (QVM) (Ouyang et al., 2022) | - | 42.5% |
| Llama-2-7b (0-shot) | Skywork-Math(Zeng et al., 2024) | - | Best-of-$N$ (QVM) | 46.8% |
| Llama-2-7b (0-shot) | Skywork-Math(Zeng et al., 2024) | - | MCTS (QVM) | 48.6% |
| Llama-2-7b (0-shot) | Skywork-Math(Zeng et al., 2024) | - | Q* (QVM) | **49.1**% |
| DeepSeek-Math-7b-RL (0-shot) | Unknown | GRPO (QVM) (Shao et al., 2024) | - | 52.1% |
| DeepSeek-Math-7b-RL (0-shot) | Unknown | GRPO (QVM) (Shao et al., 2024) | Best-of-$N$ (QVM) | 54.3% |
| DeepSeek-Math-7b-RL (0-shot) | Unknown | GRPO (QVM) (Shao et al., 2024) | MCTS (QVM) | 54.8% |
| DeepSeek-Math-7b-RL (0-shot) | Unknown | GRPO (QVM) (Shao et al., 2024) | Q* (QVM) | **55.4**% |

Table 5: Comparison of Q* and other baselines on MBPP dataset.

| Base Model | SFT | Post-training | Planning | Accuracy |
|---|---|---|---|---|
| GPT-3.5 Turbo (self-debug) (Chen et al., 2023) | Unknown | PPO (RM) (Ouyang et al., 2022) | - | 72.8% |
| GPT-4 (self-debug) (Chen et al., 2023) | Unknown | PPO (RM) (Ouyang et al., 2022) | - | **80.2**% |
| CodeGemma-7b (CodeGemma Team et al., 2024) | Unknown | PPO (RM) (Ouyang et al., 2022) | - | 65.1% |
| CodeLlama-7b (Roziere et al., 2023) | Unknown | - | - | 59.5% |
| DeepSeek-Coder-7B-Instruct-v1.5 (Guo et al., 2024) | Unknown | - | - | 75.2% |
| CodeQwen1.5-7b-Chat (0-shot) | Unknown | PPO (QVM) (Ouyang et al., 2022) | - | 74.6% |
| CodeQwen1.5-7b-Chat (0-shot) | Unknown | PPO (QVM) (Ouyang et al., 2022) | Best-of-$N$ (QVM) | 75.4% |
| CodeQwen1.5-7b-Chat (0-shot) | Unknown | PPO (QVM) (Ouyang et al., 2022) | MCTS (QVM) | 76.6% |
| CodeQwen1.5-7b-Chat (0-shot) | Unknown | PPO (QVM) (Ouyang et al., 2022) | Q* (PRM+QVM) | **77.0**% |

reasoning step. With PRM and QVM in hand, traditional methods tend to treat either of them as a verifier to select the Best-of-$N$ trajectory (Lightman et al., 2023) or utilize them to perform PPO training of RLHF (Ouyang et al., 2022). As the results shown in Table 3, we can find that with the same PRM/QVM, using it as a verifier can significantly outperform using it for PPO training in alignment. Further, in the comparison of planning-based methods, we can find that with the same QVM, Q* method with constant aggregated utility can still outperform Best-of-$N$ method. With the PRM trained on PRM800K determining whether the intermediate reasoning steps are correct, Q* method that combines PRM and QVM achieves the best performance among all methods based on the same LLM, helping Llama-2-7b surpass the performance of close-sourced ChatGPT-turbo (Shridhar et al., 2023) and reaching an accuracy of 80.8% on GSM8K dataset.

**MATH.** As the results shown in Table 4, considering the weak performance of Llama-2-7b fine-tuned with MetaMath on the MATH dataset, we seek for two other stronger LLMs to evaluate the effectiveness of Q* method. One is Llama-2-7b fine-tuned on Skywork-Math dataset (Zeng et al., 2024), which is constructed following the instruction of scaling up the SFT data, and achieves 41.9% accuracy on MATH dataset, approaching the performance of GPT-4 (Bubeck et al., 2023). The other base model is DeepSeek-Math-7b-RL (Shao et al., 2024), which could be one of the most powerful open-source 7b model for math reasoning on MATH dataset, achieving 52.1% accuracy in our reproduction. From the results shown in the second and third blocks of Table 4, we can find that Q* can still lead to further performance improvement compared to the Best-of-$N$ method on either of base models. Additionally, it is noteworthy that the performance of DeepSeek-Math-7b-RL enhanced with Q* has already surpassed a series of closed-source models on the leaderboard of MATH dataset [1], such as Gemini Ultra (4-shot) (Team et al., 2023), reaching an accuracy of 55.4% .

**MBPP.** As for MBPP dataset, we also choose one of most powerful open-source LLMs in the aspect of code generation, specifically CodeQwen1.5-7b-Chat, as our base model for evaluating the effectiveness of Q*. Following a similar procedure of math reasoning, we train a QVM for Q-value estimation and manually construct the utility function as described in the previous part of implementation details. From the results shown in Table 5, we can find that Q* can still outperform Best-of-$N$ method in the aspect of code generation, and help CodeQwen1.5-7b-Chat to achieve 77.0% accuracy on MBPP dataset, which is also a promising performance in the leaderboard of MPBB [2].

---

[1] https://paperswithcode.com/sota/math-word-problem-solving-on-math
[2] https://paperswithcode.com/sota/code-generation-on-mbpp

## 5.4 VERSATILITY OF Q*

In this subsection, we propose to demonstrate Q*'s versatility on Llama-3.1-8b (Dubey et al., 2024), showing that LLMs can leverage plug-and-play QVMs to solve various tasks using A* planning without compromising their performance on other tasks, as shown in Fig. 3. With greedy decoding, Llama-3.1-8b performs poorly, solving only 36.2% and 46.2% of problems in the GSM8K and MBPP datasets, respectively. This underperformance is unsurprising, as the one-off auto-regressive token generation process offers no opportunity for response revision. While fine-tuning on the MetaMath dataset can greatly improve performance on math reasoning problems, Llama-3.1+MetaMath performs extremely poorly on code generation tasks. In fact, we observed that Llama-3.1+MetaMath often

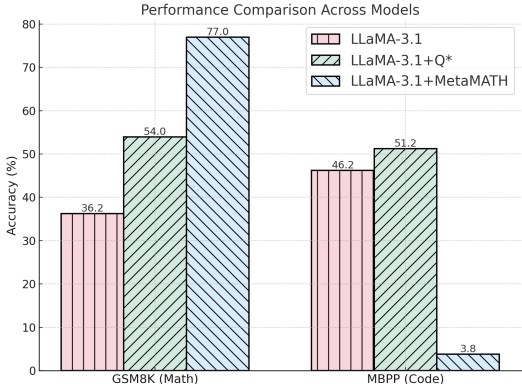

Figure 3: Performance comparison of Llama-3.1, Llama-3.1+Q*, and Llama-3.1+MetaMath.

directly introduces natural language explanations outside the comment region, resulting in faulty Python code with numerous syntax errors. In contrast, Q* substantially improves the model's performance (*i.e.,* by 17.8% on GSM8K and 5% on MBPP) by exploring the space of reasoning steps to find the most proper reasoning trajectory under the guidance of the learned QVM, eliminating the need of supervised fine-tuning and avoiding alignment tax (Askell et al., 2021; Ouyang et al., 2022). Therefore, Q* can serve as an efficient testing-time alignment method which significantly improves the performance on the targeted tasks while maintaining the model's general capabilities.

## 5.5 TESTING-TIME SCALING LAW

We examine the performance of Best-of-$N$, MCTS, and Q* on GSM8K dataset under varying decoding budgets, with results plotted in Fig. 4. Q* demonstrates the ability to refine its solution as the token budget increases, consistently outperforming Best-of-$N$. The latter, unable to provide guidance for intermediate steps during inference, requires significantly more trajectory rollouts to find the correct solution, thus consuming a large number of tokens. In contrast, Q* plans for each intermediate step, achieving superior performance even with a small token budget. MCTS, on the other hand, needs to perform costly rollout to

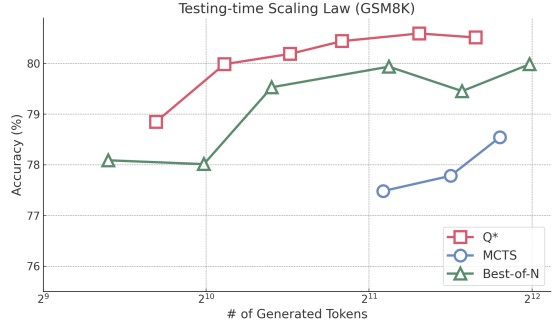

Figure 4: Testing-time scaling laws of Best-of-N, MCTS, and Q* on GSM8K dataset.

produce complete trajectories in the simulation phase of each iteration, requiring a significant amount of tokens. Moreover, the value of a state in MCTS is considered confident enough only when the state has been visited a sufficient number of times, which further exacerbates the issue.

## 6 CONCLUSION

In this paper, we present Q*, a general, versatile and agile deliberation framework for LLMs. Unlike existing deliberation methods which need extensive expertise to design a utility function for each specific task, Q* relies on ground-truth solely to train value model and can be easily applied to various reasoning tasks without modification. Moreover, by leveraging plug-and-play Q-value models as the heuristic function, Q* can effectively guide LLMs to solve various tasks without fine-tuning LLMs beforehand, which avoids potential performance degeneration on other tasks. Finally, Q* is agile because we consider only a single step each time rather than complete rollouts. Extensive empirical evaluations on math reasoning and code generation tasks confirm the superiority of our method.

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
