# OpenReview forum: "Q*: Improving Multi-step Reasoning for LLMs with Deliberative Planning"
_ICLR.cc/2025/Conference — ICLR 2025 Conference Withdrawn Submission_

### Official Review · Reviewer_dJ6k · 2024-10-17

**Soundness:** 2
**Presentation:** 3
**Contribution:** 2
**Rating:** 3
**Confidence:** 4

**Summary:**

The author presents Q*, a framework for guiding the LLMs decoding process. This framework has 2 steps: 1. Training a Q-value estimator. 2. Using A* search to find the optimal state and action. The A* search contains 2 parts: 1. g(n) is the cost of the path from the start state to the current state, using an aggregation function to calculate the cost. 2. h(n) is the cost of the path from the current state to the goal state, using a Q-value estimator to calculate the cost. The experiment shows that the Q* can improve the performance of the LLMs' planning in the GSM8k, MATH, and MBPP benchmarks.

**Strengths:**

1. The paper contains enough background and related work. The authors give a clear explanation of the Q* framework.

2. Q* seems more efficient than the other methods (like MCTS).

**Weaknesses:**

1. Bad contribution statement. Most of the paper "formalizes the multi-step reasoning of LLMs as MDP" (like RAP[1]), so this is not a contribution.

2. Lack of novelty. Both Q-value estimation[2][3]  and A*[4] search are mentioned in the related work, so the Q* is not a novel framework.

3. Lack of OOD experiment. The authors claim that the Q* is a "general framework", but the experiment only shows the result in GSM8k, MATH, and MBPP benchmarks, and the Q-value estimator is trained on these benchmarks. If the Q* can only work on these benchmarks, "the existing deliberation methods" (lines 76-77)[4] are still useful, and even better than Q* in GSM8k, so adding the OOD benchmark is needed.

4. Still need domain knowledge and manual selection. The authors claim that the Q* "does not rely on domain knowledge to design the heuristic function", however, the domain knowledge is still used in the aggregation function selection and the Q-value estimator training. And even the aggregation function selection is chosen manually and deliberately.



[1] Reasoning with Language Model is Planning with World Model

[2] Monte Carlo Tree Search Boosts Reasoning via Iterative Preference Learning

[3] Step-level Value Preference Optimization for Mathematical Reasoning

[4] TOOLCHAIN*: EFFICIENT ACTION SPACE NAVIGATION IN LARGE LANGUAGE MODELS WITH A* SEARCH

**Questions:**

1. Add some OOD experiments to show the generalization of the Q* framework. (for example, livecodebench or other math/code/agent benchmarks)

2. Remove the overstatements in the paper.

3. Compare with TOOLCHAIN* in the experiment. (Including the efficiency and the performance)

4. Section 5.4 is unrelated to this paper, why did the authors add this section? (maybe LLAMA3.1 + MetaMath + Q* is a reasonable setting)

(updated, this is not the deduct point) 5. I have realized that OpenAI has a project named Q*, so I highly recommend the author change its title or it may mislead some people.

---

### Official Review · Reviewer_H3nT · 2024-10-25

**Soundness:** 3
**Presentation:** 1
**Contribution:** 2
**Rating:** 5
**Confidence:** 4

**Summary:**

The paper presents Q*, where a Q-value function is learnt offline to estimate the "value" associated to each state in a reasoning problem. These Q-values are learnt with 3 methods: offline RL, Best of K sampling and offline MCTS. Combined with the aggregated utility function which labels the instantaneous reward associated with each state, they form the reward signal in A* search framework. During test-time, the paper performs A* search with the Q* values as the heuristic, producing the correct solution.

**Strengths:**

1. The idea behind the paper is sound. By training a Q-function in an offline manner (possibly one for different task), we can deploy them during test-time to search for correct solutions without model re-training. However, I'd like to point out that this idea is not that novel because prior works in offline RL has used the same approach (and even demonstrated on LLMs).
2. Section 5.4 demonstrates that the paper's idea is preferable over fine-tuning the model over a specific task, because fine-tuning over a specific task might cause it to do worse in other tasks.
3. The paper acknowledges that its method raises the performance of smaller model but cannot outperform GPT-4 variants (Table 3,4,5).

**Weaknesses:**

There are several weaknesses in this paper.

1. The first issue comes in the form of difference between offline data and test-time data distributions. The idea of learning a Q-function from offline data has been well studied in RL problems. However, it is well agreed in the RL domain that the problems faced during test-time might be different from those trained offline. For example, in Go and Chess, we cannot simply learn the Q-value function entirely from offline data (due to the enormous number of state space). Instead, we need to perform some variant of test-time search to find the best moves. Similarly, one might be faced with an entirely unseen math question during test-time and the Q-value, learnt from a different set of questions, is a poor heuristic during test-time. Could the authors comment about this?

2. The concept of aggregated utility g(st) is not clearly explained. I hope the author can explain clearly how the utility is learnt from static data because it forms the first half of the search heuristic (the second half uses Q-values). Furthermore, it seems difficult to learn the instantaneous utility associated with each state (e.g., a segment of code). The authors presented some ways to do so in the actual experiments, but do not explain how it is learnt clearly from offline data.

3. There are no error bars or repeated trials for experiments. Since this paper is mostly empirical, I think it is necessary to expect repeated trials and error plots.

4. There are quite a lot of grammatical mistakes and writing issues in the paper. I have marked some of them out in the next section.

**Questions:**

Some suggestions on writing and grammatical mistakes (I did not mark them all):

1. "leveraging plug-and-play Q-value models as heuristic function ..." -> "as a heuristic function", or use singular form throughout.

2. "can effectively solve various tasks via guiding LLMs to select..." -> by guiding LLMs

3. "We conduct extensive experiments on math reasoning and code generation tasks, which demonstrates ..." -> demonstrate

4. "Moreover, planning with MCTS often requires to perform costly rollout, which can significantly slow down the overall decoding process. In contrast, Q* solely relies on training a Q-value model to guide LLMs to select the most ..." -> I believe even for MCTS, a Q-value model or Q-table is learnt to guide the action selection after all the rollouts (e.g., in alphaGo). The writing needs to be refined here to reflect that the key difference is that performing MCTS during test-time is costly, while Q* learns the Q-value _offline_, and can be used directly during test-time.

5. "The solutions are Python code that is excepted ..." -> expected

---

### Official Review · Reviewer_iHBU · 2024-10-31

**Soundness:** 2
**Presentation:** 3
**Contribution:** 2
**Rating:** 5
**Confidence:** 4

**Summary:**

This paper casts the multi-step reasoning process of large language models (LLMs) as a heuristic search problem and proposes Q*, a framework based on the heuristic A* algorithm that incorporates Q-value evaluation to estimate expected future rewards. Q* aims to guide LLMs in selecting promising next reasoning step without requiring fine-tuning for the targeted task.

**Strengths:**

1.Clarity
The paper is well-structured and written in a clear, accessible way. The authors employ several advanced methods, presented logically and effectively, enhancing the reader's understanding of complex techniques.

2.Motivation
The motivation for this research is both evident and well-founded. Developing an effective, reliable framework for multi-step reasoning is a significant challenge in language model research.

3.Originality
The originality of this work is noteworthy. Few studies have investigated integrating heuristic search into LLMs to improve multi-step reasoning. By introducing heuristic search methodologies into LLMs, this paper provides a novel approach with strong experimental results.

**Weaknesses:**

1.Novelty: This work appears to be an incremental advancement based on the Reasoning as Planning (RAP) framework utilizing Monte Carlo Tree Search (MCTS). The primary contributions are as follows:

(1) Integration of the A* Algorithm: The authors incorporate the A* algorithm as the foundational framework for path searching by defining appropriate evaluation functions to guide the search process.

(2) Modification of the Heuristic Function: The traditional heuristic function h(n) in A* is replaced with Q-values. The authors employ three distinct methods to evaluate these Q-values, as detailed in Section 4.1.
While these contributions extend the existing RAP-MCTS framework, it remains some problems as below:

2.Choice of A* Algorithm
The paper utilizes the A* algorithm for heuristic search within the proposed framework. However, alternative heuristic algorithms such as Particle Swarm Optimization (PSO) [1] or Ant Colony Optimization (ACO) [2] could also be considered for heuristic search. The authors have not provided sufficient theoretical justification or empirical evidence to demonstrate that A* is the optimal choice for the path search framework in this context. Although line 194 mentions, "When the heuristic h(⋅) is admissible [3], A* guarantees to find the optimal path," this statement does not address why A* was chosen over other heuristic methods or provide comparative analysis to support its selection since no any guarantee about h(st) is admissable.

3.Reward Function Design
The design of the reward function, as described in line 240, raises concerns regarding its ability to ensure that the heuristic h(⋅) is admissible. For the A* algorithm to guarantee the optimal reasoning path, the heuristic must not overestimate the true cost from the current state to the goal. The current reward function design does not convincingly demonstrate that h(st) meets the admissibility condition, thereby casting doubt on the claim that the heuristic search can reliably produce optimal reasoning paths as stated in the authors' contributions.

4.Additional Experimental Evaluations
The paper would benefit from additional experiments to evaluate the scalability and performance of the Q* framework in larger-scale search and optimization problems. Specifically:

(1) Scalability of Q*: An assessment of how the Q* framework performs as the complexity and size of the reasoning tasks increase would provide valuable insights into its practical applicability.
(2) Comparison with Other Heuristic Algorithms: Including comparative experiments with algorithms such as PSO or ACO would help determine the relative strengths and weaknesses of using A* within this context. Such comparisons could validate the choice of A* and highlight any advantages or limitations of the Q* framework relative to other heuristic search methods.

Reference:

[1]Dorigo M, Maniezzo V, Colorni A. Ant system: optimization by a colony of cooperating agents[J]. IEEE transactions on systems, man, and cybernetics, part b (cybernetics), 1996, 26(1): 29-41.

[2]Kennedy J, Eberhart R. Particle swarm optimization[C]//Proceedings of ICNN'95-international conference on neural networks. ieee, 1995, 4: 1942-1948.

[3]Russell S J, Norvig P. Artificial intelligence: a modern approach[M]. Pearson, 2016.

**Questions:**

Please check weakness above especially the second point and the third point.

---

### Official Review · Reviewer_vjnV · 2024-11-09

**Soundness:** 2
**Presentation:** 3
**Contribution:** 3
**Rating:** 3
**Confidence:** 3

**Summary:**

The paper proposes a framework (called Q*) that guides LLM decoding toward better final solutions using an estimated optimal Q-value function. If the Q-value function is known a priori (somehow), then the method can optimize (search) solutions during test time without the need for fine-tuning. The proposed algorithm appears quite interesting at first glance for reasoning tasks. However, arguably, the formulation presented in the paper does not necessarily require planning (there is no state change/feedback for any intermediate step). Theoretically, the problem can be formulated as a single action prediction a = a_{1:T}.

**Strengths:**

- The paper is well written with clear figures that explain the algorithm and process.
- The algorithm is conceptually interesting, and the comparison of three different techniques to estimate the Q-value function is valuable.

**Weaknesses:**

- In my opinion, the major limitation is that the algorithm is conceptually designed for a different set of problems than those presented in the formulation (section 3.1) and experiments.
- The presented formulation starts with state s (question) and appends actions (autoregressively) until reaching the terminal state. There are no new observations/states or rewards for the intermediate states. The problem can be cast as a single-step process where, given a question, one needs to search over a large action space. I would argue that there is no need for multi-step reasoning.
- The presented algorithm is interesting, and I would see its value in applying it to multi-turn processes where, after applying an action (sequence of tokens), the system provides a new state, and then, conditioned on this new state/observation, one can take the next action.
- I see this work as aligning more with a beam search approach on how to get better output from an LLM when conditioning on the Q-value.
- In your experiments, could you please add uncertainty values (given that LLMs are quite stochastic)? Do the current values represent the mean or the best run?

**Questions:**

- In Algorithm 1, for the first iteration of the while loop, what would s be in line 3? The unvisited set only has the question, so what would the argmax over q result in? Perhaps you meant to initialize the unvisited set as the set of all the states in line 2?

- Similarly in line 3, if Q(s') were used instead of f(s'), how would the solution change? In the traversed path so far, g(s') remains the same for the upcoming paths and therefore doesn't influence the argmax. Thus, I think the solution would remain the same whether you use f(s'), or either of Q(s'), h(s').

---

### Note · Authors · 2024-11-30

**Comment:**

We sincerely appreciate the reviewer’s constructive feedback and are committed to improving the quality of our paper.

**Withdrawal Confirmation:**

I have read and agree with the venue's withdrawal policy on behalf of myself and my co-authors.